# Acute Effects of New Zealand Blackcurrant Extract on Cycling Time-Trial Are Performance Dependent in Endurance-Trained Cyclists: A Home-Based Study

**DOI:** 10.3390/sports11050093

**Published:** 2023-04-24

**Authors:** Stefano Montanari, Sam D. Blacker, Mark E. T. Willems

**Affiliations:** Institute of Sport, Nursing and Allied Health, College Lane, University of Chichester, Chichester PO19 6PE, UK; stefano.montanari11@gmail.com (S.M.); s.blacker@chi.ac.uk (S.D.B.)

**Keywords:** anthocyanins, sports nutrition, supplement, exercise physiology, cycling performance

## Abstract

The intake of anthocyanin-rich New Zealand blackcurrant (NZBC) extract (300 mg per day) over a week enhanced 16.1 km cycling time trial (TT) performance in endurance-trained cyclists without acute performance effects. In the present study, the acute effects of an intake of 900 mg of NZBC extract 2 h before performing the 16.1 km cycling TT were examined. A total of 34 cyclists (26 males; 8 females) (age: 38 ± 7 years, $\dot{V}$O_2max_: 57 ± 5 mL·kg^−1^·min^−1^) completed 4 16.1 km TTs (2 familiarization and 2 experimental trials) over 4 mornings on a home turbo-trainer connected with the online training simulator ZWIFT. There was no difference in time to complete the 16.1 km TT between conditions (placebo: 1422 ± 104 s; NZBC extract: 1414 ± 93 s, *p* = 0.07). However, when participants were split between faster (<1400 s; 1 female; 16 males) and slower (>1400 s; 7 females; 10 males) cyclists based on average familiarization TTs, a difference in TT performance was observed only in the slower group (placebo: 1499 ± 91 s; NZBC extract: 1479 ± 83 s, *p* = 0.02). At 12 km (quartile analysis), power output (*p* = 0.04) and speed (*p* = 0.04) were higher compared to the placebo with no effects on heart rate and cadence. The acute effects of 900 mg of NZBC extract on a 16.1 km cycling time-trial may depend on the performance ability of male endurance-trained cyclists. More work is needed to address whether there is a sex-specific time-trial effect of NZBC extract independent of performance ability.

## 1. Introduction

The application and mechanisms of fruit-derived polyphenol supplementation is an emerging field of research for sport and exercise nutrition (for a review see Bowtell and Kelly [1]). Attention has been focused on specific food sources, due to the variety of polyphenols present in fruit and vegetables. For example, an intake of cherries has provided meaningful exercise recovery effects [2]. Blackcurrant is an anthocyanin-rich food source with an average content of 250 mg for 100 g of fresh fruit [3]. Anthocyanins are water soluble compounds and the largest subcategory of polyphenols and flavonoids with antioxidant, anti-inflammatory, and vasoactive properties [4,5]. The blackcurrant anthocyanin cyanidin–3–glucoside, for example, facilitates nitric oxide synthase expression in bovine artery endothelial cells, increasing the bioavailability of the vasodilator nitric oxide [6]. In addition, a 7 day intake of anthocyanin-rich NZBC increased cardiac output and reduced total peripheral resistance in endurance-trained female and male cyclists at rest [7]; it enhanced 16.1 km cycling performance in male endurance-trained cyclists [8] and high-intensity running performance in recreationally active males [9]. Beneficial effects with a 7 day intake of NZBC are likely due to the bioavailability of anthocyanin-derived metabolites causing cellular adaptations. However, the observed performance effects may be driven by the raise in blood plasma anthocyanin and anthocyanin-derived metabolites after the consumption of the last dose before the trial. Czank et al. [10] showed a peaking of plasma cyanidin–3–glucoside approximately 2 h post intake with the recovery of many anthocyanin-derived metabolites in plasma, urine, faeces, and breath over a period of 48 h after an intake of 500 mg of cyanidin–3–glucoside. With an intake of capsulated 300 mg of NZBC extract (~105 mg anthocyanins), the presence of the second phase metabolites protocatechuic acid and gallic acid were observed 1.5 and 4 h after [11].

There is evidence of the acute intake of blackcurrant anthocyanins having beneficial effects on vascular function [4] and reducing oxidative stress [12]. However, the acute effects of blackcurrant anthocyanins on exercise performance (i.e., taking a supplement 2 h before exercise) have not been extensively addressed. Montanari et al. [13] observed no acute effects of 300 (one capsule) and 600 mg (two capsules) of NZBC extract on the 16.1 km cycling time-trial performance in laboratory tests. In addition, there were no acute effects of blackcurrant intake on repeated (8 × 5 min) maximal intensity cycling bouts [14]. Similarly, Barnes et al. [15] did not observe a difference in the number of maximal isometric grip contractions after the acute intake of a 1.87 mg/kg body weight of blackcurrant anthocyanins. The doses used in these studies may not have been sufficient to exert an acute biological effect. Rodriguez-Mateos et al. [16] demonstrated that acute anthocyanin intake enhanced forearm blood flow at rest in a dose dependent manner, reaching a plateau with 310 mg of wild blueberry anthocyanins.

As far as we know, no studies have examined exercise performance effects by a higher dose of NZBC extract than the amount used in previous studies [13]. Therefore, the aim of this present study is to examine the acute effects of 900 mg of NZBC extract (3 capsules, i.e., 315 mg of anthocyanins) (taken 2 h before) on a 16.1 km cycling time-trial performance.

## 2. Materials and Methods

### 2.1. Participants

Recruitment for this study was during COVID-19, restricting laboratory access and testing. Therefore, participants were recruited with access at home to a cycling turbo-trainer with power recording and the online cycling exercise and training software ZWIFT (Amsterdam, The Netherlands). Sixty endurance-trained male and female cyclists were recruited. Participants had to be healthy, not involved in a structured physical training program, and have a chest strap for heart rate recording. Drop-out was due to illness (*n* = 8), injury (*n* = 6), turbo-trainer malfunction (*n* = 5), family reasons (*n* = 4), and relocation (*n* = 3). An amount of 26 males (age: 38 ± 7 years; height: 179 ± 6 cm; weight: 75 ± 7 kg, $\dot{V}$O_2max_: 62 ± 8 mL·kg^−1^·min^−1^, W_max_: 531 ± 133 W) and 8 females (age: 35 ± 10 years; height: 166 ± 9 cm; weight: 61 ± 7 kg, $\dot{V}$O_2max_: 57 ± 5 mL·kg^−1^·min^−1^, W_max_: 316 ± 98 W) completed the study. The $\dot{V}$O_2max_ was obtained by participants providing information from individually owned wearable devices. Participants provided written informed consent and completed a food frequency questionnaire for the calculation of total anthocyanin intake (45 ± 18 mg·day^−1^) using the phenol explorer database [17].

### 2.2. Experimental Design

The study adopted a randomized (http://www.randomization.com, accessed on 6 April 2023), double-blind, and placebo-controlled crossover design. Participants were informed that the study examined the effects of two different NZBC extract supplements. Participants completed 4 morning testing sessions (±2 h) within a month. For each session, participants fitted a heart rate monitor and connected via Bluetooth to the ZWIFT online platform. Each session had a 20 min warm-up at 65% of their estimated functional threshold power, followed by a 10 min rest before completing the 16.1 km cycling time trial. On completion, participants recorded a rating of perceived exertion (6–20 scale) [18]. The first two sessions were familiarization TTs to allow the calculation of the coefficient of variation (CV%). The 2 experimental sessions were separated by a minimum of 5 and maximum of 10 days to allow wash-out [10].

### 2.3. Online Software and Data Extraction

Participants connected the turbo-trainers with the online training software ZWIFT (https://www.zwift.com, accessed on 6 April 2023) (Amsterdam, The Netherlands). ZWIFT is a virtual platform, with more than 2.5 million registered users. A personal account allowed the same time-trial bike, excluding drafting and similar self-selected virtual bike settings (wheels, frame, and tires). Participants completed the Tempus Fugit track in the WATOPIA region. A total of 1 lap is approximately 20 km with only 16 m elevation. On the completion of the warm-up, participants reset the map and started the 16.1 km time trial from a still position. Data files (“.fit” format) were emailed to the investigator (SM) for analysis using the GoldenCheetah software version 3.5. Time, power, speed, distance, cadence, and heart rate were recorded at 1 s intervals.

### 2.4. Menstrual Cycle Monitoring

Females completed a menstrual cycle questionnaire to monitor the participants’ symptoms during the experimental trials. Participants reported on oral contraceptives and stage in the menstrual cycle. The prevalence of emotional and physical symptoms from 1 (never) to 5 (very often) were rated (symptoms adapted from Martin et al. [19]).

### 2.5. Diet Standardization and Supplementation

Before each test, participants avoided strenuous exercise for 48 h, did no exercise, and had no alcohol intake for 24 h. Participants were instructed to record food intake for 24 h before testing and replicate for subsequent tests. Breakfast was consumed 3 h before starting the warm-up, with no restriction on caffeine intake. Thereafter, only water intake was allowed. Energy and fluid intake and macronutrients were quantified (Nutritics v5, Dublin, Ireland). For the 2 experimental tests, participants consumed 900 mg of NZBC extract (3 capsules) or a placebo (PLA, microcrystalline cellulose M102) in a randomized order. Each NZBC extract capsule contained 105 mg of anthocyanins (i.e., 35–50% delphinidin–3–rutinoside, 5–20% delphinidin–3–glucoside, 30–45% cyanidin–3–rutinoside, and 3–10% cyanidin–3–glucoside) with the remaining primarily natural sugars (CurraNZ™, Health Currancy Ltd., Surrey, UK). Capsules were taken 1.5 h before starting the warm-up. Intake timing was based on blood peak anthocyanin concentration approximately 2 h post consumption [10] to coincide with the TT start.

### 2.6. Statistical Analysis

JASP 0.14.10. (jasp-stats.org) was used. Normal distribution was assessed with the Shapiro–Wilk test. Paired sample *t*-tests were used to compare time, power, speed, heart rate, and cadence between conditions. Repeated measures ANOVA using a condition (PLA vs. 900 mg·day^−1^ of NZBC extract) by distance design (i.e., 4, 8, 12, and 16.1 km) was implemented to investigate the main effects on distance, condition, and interaction with post-hoc Bonferroni correction. *T*-test analysis determined if there was a difference between conditions at single time points and in fast and slow cyclists based on individual mean time from the two familiarization trials in the cohort. For example, participants with an individual average time slower than the mean time of the cohort of the familiarization trials were considered slow cyclists in the cohort. Data were checked for order effects using a mixed model. Results are reported as mean ± SD with the mean difference and 95% CI. Data from the familiarization trials were averaged to determine the coefficient of variation (CV), the typical error (TE), and the smallest worthwhile change (SWC). SWC was calculated multiplying the SD by 0.5 [20]. Cohen’s *d* effect size was interpreted as trivial (0 ≤ *d* < 0.2), small (0.2 ≤ *d* < 0.5), medium (0.5 ≤ *d* ≤ 0.79), and large (*d* ≥ 0.8). Power analysis indicated that a sample size of 50 would allow the detection of a small effect size (*d* = 0.4) for TT performance with a high statistical power (1 − β = 0.80; 0.05 = α level). Significance was set at *p* ≤ 0.05.

## 3. Results

### 3.1. Diet and Menstrual Cycle Symptoms

There were no differences for calorie intake (PLA: 2144 ± 351; NZBC: 2200 ± 408 kcal), carbohydrates (PLA: 258 ± 73; NZBC: 260 ± 82 g), fat (PLA: 75 ± 23; NZBC: 76 ± 26 g), and protein (PLA: 113 ± 30; NZBC: 107 ± 29 g). At breakfast, 17 participants had an average caffeine intake of 125 ± 68 mg. The menstrual cycle questionnaire provided no difference between conditions for any of the symptoms (Table 1).

### 3.2. 16.1 km Performance—Cohort Observations

The CV for the two familiarization trials was 1.26% with a SWC of 12.2 s and a TE of 17.1 s. With NZBC extract, the change in TT did not meet the conventional level of statistical significance (PLA: 1422 ± 104 s, 95% CI [1386, 1459 s]; NZBC: 1414 ± 93 s, 95% CI [1382, 1449 s]; *p* = 0.07) and had a trivial effect size (*d* = −0.08) (Table 2). No differences were observed for power, speed, heart rate, cadence, and RPE (Table 2). There was no interaction between the allocation of the condition and the time to complete the trials (*p* = 0.45).

### 3.3. 16.1 km Performance—Slow and Fast Cyclists

The individual average time from the familiarization trials allowed a division of the cohort into fast (TT < 1400 s) and slow cyclists (TT > 1400 s). There was no significant correlation between habitual anthocyanin intake and TT performance. Slower cyclists (7 females and 10 males) had a CV of 1.31% (TE = 20 s) and faster cyclists (1 female and 16 males) a CV of 1.20% (TE = 13 s) with a SWC of 14.3 and 9.2 s. For the slow cyclists, 16.1 km time was 20 s faster with NZBC (PLA: 1499 ± 91 s, 95% CI [1452, 1546 s]; NZBC: 1479 ± 83 s, 95% CI [1437, 1522 s]; *p* = 0.02) with a small effect size (d = −0.23). For the slow cyclists, higher cycling power and speed were observed with NZBC but without an effect on HR and cadence (Table 3). For the fast cyclists, there were no differences in cycling time (PLA: 1345 ± 40 s, 95% CI [1325, 1367 s]; NZBC: 1349 ± 43 s, 95% CI [1326, 1371 s], *p* = 0.34), power, speed, heart rate, and cadence.

Figure 1a,b shows the 16.1 km TT data for each condition, the individual times, and the plot with the mean difference with 95% CI for the slower and faster cyclists.

Figure 2 shows the relationship between the placebo time in the 16.1 km TT and the percentual change in the 16.1 km TT with the NZBC extract condition for all participants. The significance of the negative correlation (*p* = 0.02) indicates that slower cyclists may have benefitted a performance-enhancing effect by the intake of NZBC extract.

For the slow cyclists, there was no difference between time points, conditions, and interaction in time to complete the 16.1 km TT (Figure 3a). Power data analysis showed a difference in condition (F = 5.22, *p* = 0.03) and distance (F = 4.16, *p* = 0.01) with no interaction. Specifically, the power profile across the whole TT in the slow cyclists was higher with NZBC and close to conventional significance at 8 km (PLA: 234 ± 54, NZBC: 239 ± 54 W, *p* = 0.06, *d* = 0.09) and different at 12 km (PLA: 232 ± 54, NZBC: 239 ± 51 W, *p* = 0.02, *d* = 0.13) (Figure 3b). Similarly, there was a difference in speed (F = 4.85, *p* = 0.04) with data close to conventional significance at 8 km (PLA: 38.9 ± 2.5, NZBC: 39.3 ± 2.5 km·h^−1^, *p* = 0.07, *d* = 0.16) and different at 12 km (PLA: 38.9 ± 2.5, NZBC: 39.3 ± 2.5 km·h^−1^, *p* = 0.02, d = 0.16) (Figure 3c). Cadence and heart rate showed a difference for distance (F = 79, *p* < 0.001; F = 5.22, *p* = 0.003) with no difference for condition or interaction (Figure 3d,e). For the 4 km split for the fast cyclists, there was no difference between conditions, but a difference in distance for time (F = 27.9, *p* < 0.001, Figure 3f), power (F = 4.4, *p* = 0.008, Figure 3g), speed (F = 28.7, *p* < 0.001, Figure 3h), cadence (F = 27.9, *p* < 0.001, Figure 3i), and heart rate (F = 186.9, *p* < 0.001, Figure 3j).

## 4. Discussion

We provided novel observations using a home-based study design on the acute effects of an anthocyanin-rich supplement (900 mg of extract with 315 mg of anthocyanins) on the 16.1 km cycling performance in endurance-trained females and males. For the cohort, no acute effect of NZBC extract on TT performance was observed. However, in the 34 participants of the cohort, 63% of the females (5/8) and 35% of the males (9/26) cyclists had an improvement of >0.6%, considered the smallest worthwhile change in elite cyclists in road time-trials [20]. It would be of interest to examine whether such a response is repeatable and whether there is a sex-specific response. However, Best et al. [21] provided an absence of a performance-enhancing effect in females with a repeated sprint exercise with a 7 days intake of NZBC extract, suggesting that a sex-specific response may depend on exercise modality. In the present study, when participants were split between fast (<1400 s) and slow (>1400 s) cyclists, time to complete the 16.1 km TT was enhanced in the slower cyclists with NZBC extract with 11 from 17 showing an improvement of >0.6%. The faster 16.1 km TT in the slower group was determined by a higher average power and speed profile across the whole TT, being close to significance at 8 km and different at 12 km with no difference for the faster group. Although our observations suggest a benefit of acute NZBC intake on high-intensity cycling endurance performance in slower cyclists, caution is warranted considering the strength and limitations of the present study design. Prior power analysis suggested that a sample size of 50 cyclists was needed to determine a small effect size with high statistical power. However, only 34 cyclists completed the study.

Several reasons might explain the difference in the slower and faster cyclists. First, during high-intensity cycling, the increase in blood flow and shear stress promotes endothelium-dependent vasodilatation via the upregulation of endothelial nitric oxide synthase (eNOS). Anthocyanin can increase nitric oxide (NO) bioavailability through enhanced eNOS activity; however, human data are scarce and contradictory. For example, the acute intake of 1.87 mg·kg^−1^ of blackcurrant anthocyanins did not change NO levels after 2 h in healthy individuals [15]. Similarly, 60 mL of Montmorency cherry (MC) did not increase plasma NO in trained cyclists [22]. Moreover, endurance training increases NOS expression [23]. Therefore, it may be possible that NO production was already optimal in the most well-trained participants in our cohort. Future studies should address whether an acute intake of NZBC extract can affect NO bioavailability pre- and post-high-intensity exercise in cyclists and if it correlates with cycling endurance performance. In contrast to our study, Keane et al. [22] observed 10% higher total work completed over a 60 s all out cycling sprint after consuming a single dose of 60 mL MC concentrate, providing ~145 mg of polyphenols and ~65 mg of anthocyanins. Interestingly, they did not observe changes for nitrogen dioxide or nitrate in the MC-supplemented trial when compared to a placebo. It is possible that the antioxidant effects of polyphenols in the MC concentrate might have mitigated the rise in reactive oxygen species (ROS) during high-intensity exercise favoring muscle contraction [24].

Nuclear factor–erythroid–factor–2–related factor 2 (Nrf2) is a transcription factor that has a key role in protecting the cell via the upregulation of antioxidant enzymes in response to oxidative stress. A recent study showed that 4 weeks of high-intensity training increased Nrf2 in skeletal muscles by 1.6 ± 0.5-fold [25]. Therefore, faster cyclists could have had better training adaptation with a higher endogenous upregulation of Nrf2 and antioxidant defenses which could not be further enhanced by an acute intake of NZBC extract.

Few studies have examined the acute effects of anthocyanin intake on exercise performance with mixed results. For example, the acute consumption of 60 mL of Montmorency cherry increased peak power output and total work completed by 9.5% and 10% in sprint cycling [22]. The intake of a single dose of blackcurrant (1.87 mg·kg^−1^ body weight) mitigated the decline in forearm blood flow during 120 min of prolonged sitting but did not increase the number of isometric hand-grip contractions in healthy males [15]. Montanari et al. [13] observed that the acute intake of 300 mg and 600 mg of NZBC extract had no effect on 16.1 km TT performance. In our study, the higher dose (900 mg extract with 315 mg of anthocyanins) provided a performance-enhancing effect in the slower cyclists. It is possible that in the slower cyclists, a different oxidative stress response compared to the faster cyclists allowed the effectiveness of the New Zealand blackcurrant extract to enhance muscle contraction during the time-trial.

Second, another reason might be related to the experience of the cyclists. Due to the home-based nature of the study, participants had data feedback from the screen. Therefore, they had full knowledge of their performance in real time. The pacing strategy during a race plays a key role in determining the performance outcome [26]. For an event longer than >4 min, there is evidence that an even distribution of energy is optimal for performance [27,28]. In the present study, both the slow and fast cyclists averaged the highest power output in the first 4 km; however, only the faster cyclists were able to produce a constant power output from the 8 km mark until the end of the race. This is also supported by the lower variation in the faster group. Average standard deviation for time and power was almost half compared to the slowest group (Table 3). Therefore, we cannot exclude that the faster cyclists were more experienced in using pacing strategies to optimize performance. However, future studies can additionally address whether the intake duration of NZBC extract needs to be longer in the faster male cyclists to experience ergogenic performance effects.

Although it is concluded that the acute intake of 900 mg of NZBC extract enhanced performance in the slower group, we need to consider the limitations of the present study. Firstly, this was the first home-based study examining the effects of NZBC extract on cycling endurance performance. Two contributing factors to endurance performance are the perception of effort and the potential motivation (the greatest amount of effort that a person would be willing to offer to satisfy a motive) [29]. We asked the participants to provide their RPE at the end of each trial. There was no difference between the two conditions. However, we did not assess individual levels of motivation. Elite endurance athletes are able to plan the use of cognitive strategies (e.g., to focus on breathing) to enhance motivation and outcome expectations [30]. Future home-based study might include a questionnaire assessing the motivation levels of the participants to better understand their commitment towards the task assigned. In addition, participants were not blind to the time and distance information. We have no record of whether participants were affected by preceding TT performances in the study. Secondly, the present study includes males and females in the cohort. A recent meta-analysis has indicated that performance in females might be trivially reduced during the early follicular stage of the menstrual cycle with low quality of evidence [31]. Those who use an oral contraceptive might have a trivial reduction in performance outcomes compared to naturally menstruating women; however, no difference is observed within the oral contraceptive users’ cycle [31]. Elliott-Sale et al. [32] suggested that if the main outcome of the study design is performance, a simple screening of the menstrual status at the time of the testing should be sufficient. We therefore did not standardize the specific menstrual cycle phases in our female cyclists. However, females rated their menstrual cycle symptoms on the morning of each trial. We did not find a difference between the placebo and NZBC extract conditions. Therefore, we assume that the menstrual cycle in the sample of eight females did not have an impact on the performance in the present study. Thirdly, we need to address the variation of our study model. Paton and Hopkins [20] reported that the smallest worthwhile was 1.7% for cycling competitions between 17 and 59 km. In our study, the SWC was 14.3 s for the slower group and 9.2 s for the faster group. With NZBC extract, the slower group finished the 16.1 km TT 20 s faster. However, the CV calculated from the 2 familiarization trials was 1.31% for the slower group which resulted in a typical error of ~20 s. Therefore, it is likely that the faster time with NZBC intake in the slower group is a representation of the variation of our model. It is demonstrated that the inclusion of a familiarization session can increase the reliability of a performance test [33]. However, the number of familiarization sessions might differ depending on the experience of the cyclist. For example, if the cyclist is not accustomed to the TT format, up to 4 familiarizations might be required to bring the CV below 2% for a 20 km TT [34]. Zavorsky et al. [35] showed that the CV over a 16.1 km TT ranged from 4.8 to 1.2% with the slower cyclist showing higher variation. Therefore, it is possible that some of the cyclists in the slower group might have benefitted more from an additional familiarization session to improve their experience with the TT format. Future studies should consider customizing the number of familiarization sessions depending on the experience and fitness level of the cyclist recruited. Finally, we decided not to exclude habitual caffeine intake. In the group of slow cyclists, 6 of the 11 cyclists had an improvement of >0.6% and reported caffeine intake on the days of testing but in both conditions. However, Paton et al. [14] did not observe a synergistic effect of the combined intake of caffeine and blackcurrant in a model of repeated high-intensity cycling bouts.

## 5. Conclusions

The acute intake of 900 mg of NZBC extract did not improve the 16.1 km home-based cycling performance in a cohort of endurance cyclists (*n* = 34, 8 females). When the cohort was split into slower and faster cyclists, we observed a significant effect on the slower group with a small effect size. The slower cyclists may have benefitted from the antioxidant properties of blackcurrant anthocyanins and anthocyanin-derived metabolites to counteract the excess in ROS production during the 16.1 km TT. However, the difference in performance might have been driven by the level of experience of the cyclists. Therefore, caution is required when interpreting the data. In conclusion, 900 mg of NZBC extract is beneficial for enhancing performance in male cyclists with a lower fitness level and in females. Future studies should have sufficient participants to allow a comparison between male and female cyclists based on cycling ability.

## Figures and Tables

**Figure 1 sports-11-00093-f001:**
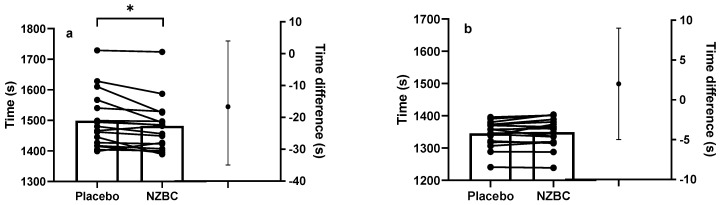
Individual 16.1 km time−trial performance and the mean group difference with 95% CI in the placebo and NZBC extract conditions for the slower (**a**) and faster (**b**) cyclists. NZBC, New Zealand blackcurrant; CI, confidence interval. *, indicates a difference between conditions.

**Figure 2 sports-11-00093-f002:**
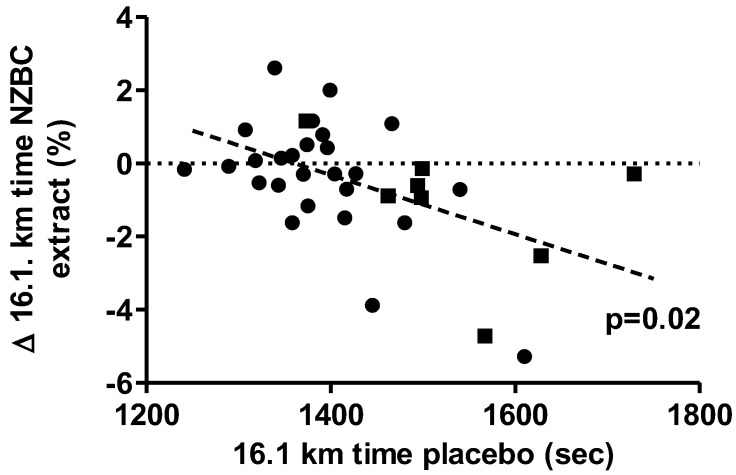
Relationship between the 16.1 km time in the placebo condition and the % change in 16.1 km time in the NZBC extract condition. NZBC, New Zealand blackcurrant. Black circles: males; black squares: females. The significant linear correlation is for the cohort.

**Figure 3 sports-11-00093-f003:**
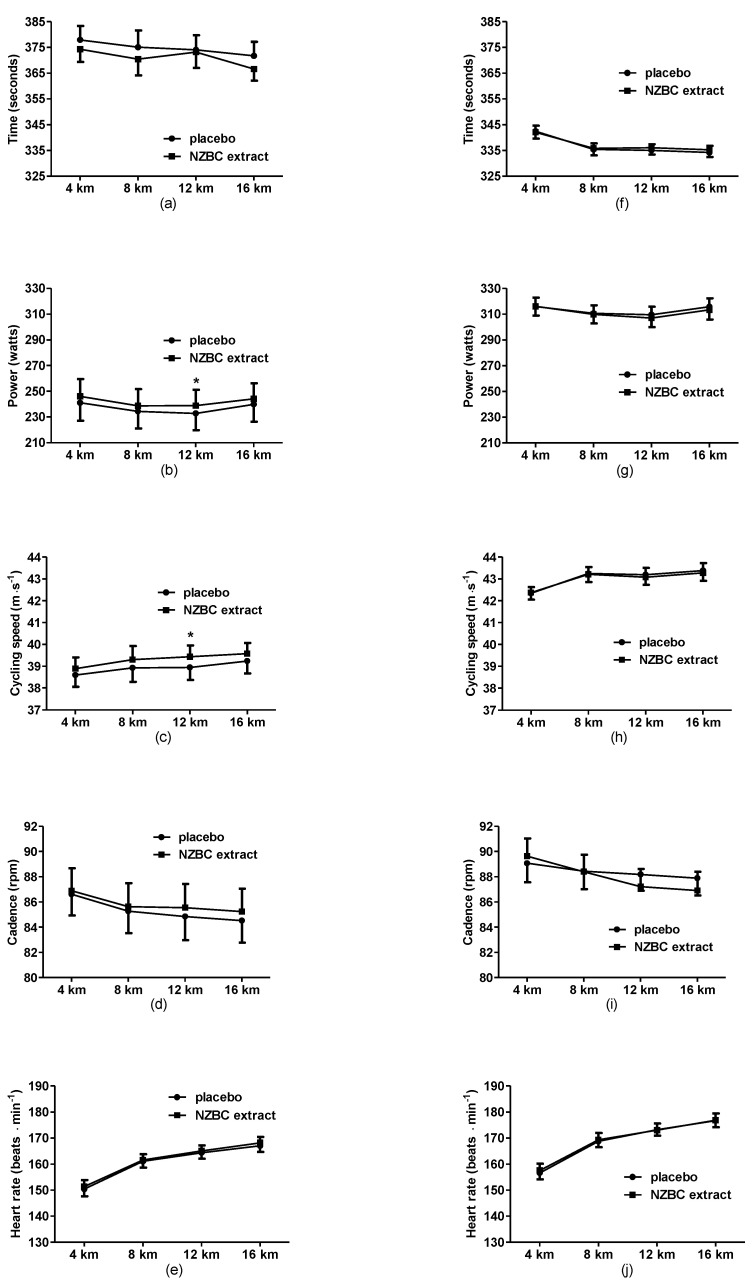
Mean data over 4 km splits of the 16.1 km time-trial for time, power, cycling speed, heart rate, and cadence in the slower cyclists (**a**–**e**) and faster cyclists (**f**–**j**). rpm, revolutions per minute. *, indicates a difference between placebo and NZBC extract conditions (*p* < 0.05). NZBC, New Zealand blackcurrant.

**Table 1 sports-11-00093-t001:** Menstrual cycle symptoms in endurance-trained female cyclists (*n* = 8).

Symptoms	Placebo	NZBC Extract	*p*-Value	Effect Size Cohen’s *d*
Stomach pain	1.1 ± 0.3	1.3 ± 0.7	0.35	0.37
Headache/migraine	1.1 ± 0.3	1.1 ± 0.1	0.68	0.00
Bloating	1.1 ± 0.4	1.6 ± 0.7	0.22	0.88
Nausea/vomiting	1.1 ± 0.4	1.6 ± 0.7	0.17	0.88
Tiredness/fatigue	1.4 ± 0.7	1.7 ± 0.4	0.55	0.53
Dizziness	1.1 ± 0.1	1.1 ± 0.3	0.68	0.00
Irritability	1.1 ± 0.1	1.4 ± 0.3	0.45	1.34
Hunger/appetite	1.2 ± 0.3	1.3 ± 0.7	0.35	0.19
GI distress	1.1 ± 0.3	1.6 ± 0.7	0.68	1.00
Heavy bleeding	1.1 ± 0.4	1.5 ± 0.7	0.17	0.70
Muscle ache	1.1 ± 0.5	1.5 ± 0.5	0.18	0.80
Weakness	1.1 ± 0.3	1.5 ± 0.4	0.54	1.13
Mood swings	1.1 ± 0.3	1.3 ± 0.5	0.35	0.49
Flustered	1.1 ± 0.3	1.1 ± 0.3	0.35	0.00

NZBC, New Zealand blackcurrant; GI, gastrointestinal. Data are mean ± SD.

**Table 2 sports-11-00093-t002:** Performance parameter for the 16.1 km time-trial (*n* = 34; 8 females).

Parameter	Placebo	NZBC Extract	MD	95% CI for MD		*p*-Value	Effect Size Cohen’s *d*
				low	upper		
Time (s)	1422 ± 104	1414 ± 93	−8 ± 25	−17	1	0.06	−0.08
Power (W)	275 ± 57	277 ± 55	2 ± 9	−1	5	0.27	0.04
Speed (km·h^−1^)	40.9 ± 2.8	41.1 ± 2.6	0.2 ± 0.7	0	0.4	0.10	0.07
Heart rate (bpm)	166 ± 10	165 ± 10	1 ± 3	0	2	0.15	−0.10
Cadence (rpm)	87 ± 6	87 ± 6	0 ± 3	−1	1	0.65	0.00
RPE	17 ± 1	17 ± 1	0 ± 1	0	0	1.00	0.00

NZBC, New Zealand blackcurrant; bpm, beats per minute; rpm, rotations per minute; RPE, rating of perceived exertion; MD, mean difference. Data are mean ± SD.

**Table 3 sports-11-00093-t003:** Performance data for the slower and faster cyclists.

Parameter	Placebo	NZBC Extract	MD	95% CI for MD		*p*-Value	Effect Size Cohen’s *d*
Slow cyclists (>1400 s, *n* = 17)				lower	upper		
Time (s)	1499 ± 91	1479 ± 83	−19 ± 30	−35	4	0.02	−0.23
Power (W)	237 ± 55	242 ± 52	5 ± 9	0	9	0.04	0.09
Speed (km·h^−1^)	38.7 ± 2.2	39.2 ± 2.0	0.4 ± 0.7	0.0	0.7	0.04	0.18
HR (bpm)	160 ± 9	161 ± 9	1 ± 4	−1	3	0.43	0.11
Cadence (rpm)	85 ± 7	86 ± 7	1 ± 3	−1	2	0.57	0.14
Fast cyclists (<1400 s, *n* = 17)							
Time (s)	1345 ± 40	1349 ± 43	2 ± 14	−5	9	0.34	0.10
Power (W)	313 ± 26	312 ± 29	−1 ± 9	−6	3	0.51	−0.04
Speed (km·h^−1^)	43.0 ± 1.2	43.0 ± 1.4	0 ± 0	0	0	0.59	0.00
HR (bpm)	169 ± 10	169 ± 10	0 ± 2	−1	2	0.53	0.00
Cadence (rpm)	88 ± 6	88 ± 6	0 ± 3	−2	1	0.58	0.00

NZBC, New Zealand blackcurrant; HR, heart rate; bpm, beats per minute; rpm, rotations per minute; MD, mean difference. Data are means ± SD.

## Data Availability

Data can be provided on reasonable request.

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
