# Peer review of "Acute Effects of New Zealand Blackcurrant Extract on Cycling Time-Trial Are Performance Dependent in Endurance-Trained Cyclists: A Home-Based Study"

_sports, 2023, doi:10.3390/sports11050093_

Round 1

Reviewer 1 Report

Thank you for the opportunity to review this paper, which is a clear and logical extension of the authors' or group's previous work. The data presented are interesting and it's good to see modern training/competition methods being employed to gather data (i.e. ZWIFT).

Please find below a section by section review.

Abstract

I am happy with the abstract, and have no recommendations at this stage.

Introduction

29 - please add a closing bracket after the reference [1]

Please review this section for possible paragraph breaks. You cover a range of themes and collate a lot of evidence in a single block of text. For readability/ ease of interpretation it is recommended to break this into two or three related paragraphs.

Methods

126 - please amend 'was' to 'were', as you've performed multiple t-tests

128 - you may need to add 'design' after distance, to further describe the ANOVA and intended interactions

130/132 - you cover two different statistical processes here. I'd like to know a little more about how you divided your participants into fast/slow groups and if this was based upon any previous criteria or work, or if you simply ranked and grouped participants' data

131 and 134 - you use the term average on both lines. Am I right to think this is a mean?

137/138 - using the thresholds you've just described the effect size you are powered to detect with n=50, would be considered small?

Results

Typically you wouldn't present a 95% CI, alongside a standard deviation of the same mean result, as both CI and SD are measures of distribution that demonstrate a likely range of observations. I think this is then further confused by only presenting the CI of the mean difference, but not presenting the mean difference itself, in Table 2.

Have you rounded most effect sizes/ to 1 decimal place? If so why, when most authors and indeed Cohen tend to report to 2 dp?

172 - please amend effect to effects

Table 3 - please check direction of your greater than sign for faster cyclists, and you may want to add another ) at the end of each sample description to encapsulate time and sample size

Figure 1a - you may wish to revise the mean difference parameter as this currently shows in the positive, indicating a greater time to completion, compared to 1b which appears correct, based on the data presented in Table 3.

Discussion

209/210 - this seems like a group as opposed to individual level comment?

222 - how does this affect your confidence in your observations? Can we be as certain, especially given small to moderate effects and breadth of confidence intervals observed?

224/276 - please consider breaking this down into more paragraphs; you also introduce some new abbreviations that readers may not be familiar with. Please write in full in the first instance.

I think your final appraisal of your findings is refreshingly honest, and you give full consideration to a range of measure that perhaps demonstrate that your participants improved, but within the anticipated error of the testing. This is to be commended.

You may wish to include the below reference which also assessed repeated sprint performance in female athletes, following NZBC supplementation. This may lend support to the idea that sex differences in response to anthocyanin consumption may occur, dependent upon exercise modality and substrate metabolism requirements

Best, R., Metekingi, C., Longhurst, G., & Maulder, P. S. New Zealand Blackcurrant extract supplementation does not improve repeated sprint ability. Age (years)25(4), 31-7.

Reviewer 2 Report

The research has some merit, given the currently limited research on Blackcurrant supplementation. The study sample size is good, and the manuscript is generally well presented (especially the introduction). However, several issues detract from the study and should be addressed. An obvious problem is the lack of scientific rigor in data collection. Studies of this kind would typically be completed in a controlled laboratory with careful environmental monitoring etc., not in the subjects' homes under potentially variable conditions (presumably non-blinded pacing conditions). Unfortunately, this is not a fixable issue. That aside, the statistical approach used is questionable, and there appear to be errors in calculating some values (particularly Cohen effect sizes which seem inflated or missing in some cases). I strongly advise that the statistical results are carefully checked for accuracy. There also appears to be a tendency of cherry-picking data to highlight potential benefits of BC where none exist (or are at least unsupported by the current findings. The discussion is also often speculative and delves into physiological concepts and explanations far beyond the study data or scope.

I have provided comments in a supplied copy document that the authors should address to improve the manuscript further.

Reviewer 3 Report

The article entitled “Acute effects of New Zealand blackcurrant extract on cycling time-trial are performance dependent in endurance trained cyclists: A home-based study” has been presented by Mark et.al. However, the article can be considered for publication, if address the following comments carefully.

1)      The author has several publications on such extract, but there is no specificity in such work, you need to elaborate well, what is the need of such study by comparing it to your previous work, also if here you study only a single factor, why it’s not part of previous work. Explain it well in the introduction and discussion portion of your study.

2)      If previously 310 mg of anthocyanin was studied already, then you need to clearly mention what will happen (expect) if you increase it to 315 mg?

3)      Mention the exact detail of your extract in the material and method portion.

4)      Based on which evidence do you assume that the menstrual cycle of 8 females have not any impact on the performance? You need to elaborate well this statement with reference.

5)      Synergistic effect of acute intake of caffeine and blackcurrant in such performance, what did you expect from this statement? 

Round 2

Reviewer 2 Report

Thank you for your updated manuscript and response to my comments. It is a substantial improvement and now provides readers with a more precise results analysis and valuable interpretation. Figure 2 provides a much better understanding of individual responses though it would be beneficial to include the r or r2 and the stated p-value (0,02). Likewise, thanks for including the mean change scores and updating Cohens' D values.

I still feel you are pushing the limits of credibility (p hacking) with the separate quadrant analysis and the reported significance at just one quadrant (12km); however, I am happy to let educated readers make their own interpretation of this reported finding. You're also missing an opportunity by not providing further interpretation and comment regarding the menstrual cycle symptoms. Your argument regarding the lack of p value significance is unconvincing. Most reputable journals now call for more emphasis on the effect's magnitude, not the p values (which many mistakenly interpret as no effect when p>0.05). Your data shows Large magnitude effects (D>0.8) for bloating, nausea and GI distress (making a consistent response pattern), and I feel this is important and worthy of comment. This is similar to the effects reported for Sodium Bicarbonate ingestion and is why it is not widely used as an ergogenic aid in endurance sports. A couple of sentences in the discussion would provide a novel finding, which prompts other researchers to investigate this further.

Reviewer 3 Report

In my previous comment no. 2, "    If previously 310 mg of anthocyanin was studied already, then you need to clearly mention what will happen (expect) if you increase it to 315 mg?", It seems that the author didn't understand it well, or may be I am confused" please clarify line  63-64 ". Rodriguez-Mateos et al demonstrated that acute anthocyanin intake enhanced forearm blood flow at rest in a dose  dependent manner reaching a plateau with 310 mg of anthocyanins. elaborate this sentence in view of comment No 2. 

Author Response

Apologies for lack of clarity in our response. One capsule of the NZBC extract (CurraNZ) contains 105 mg of anthocyanins. We did not manufacture the capsules but use commercially available capsules. 

We agree with the reviewer that it is confusing to state previous research establishing with 310 mg of anthocyanins a levelling of of the flow-mediated dilation. 

We have clarified by mentioning that 300 and 600 mg of NZBC extract was one and two capsules and that we are using now three capsules. We have also clarified that the study with 310 mg of anthocyanins were wild blueberry anthocyanins.

We hope this clarifies.

Thanks again for your time and engagement.